# MTGS: Multi-Traversal Gaussian Splatting

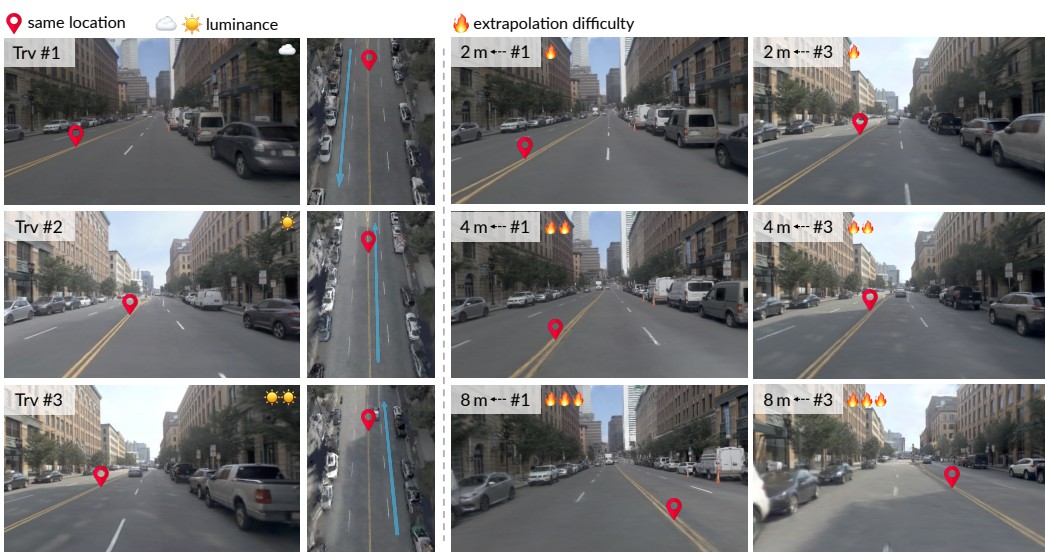

(a) Various Appearance on Ego view & Bird Eye's View 🌎    (b) View Extrapolation to Anywhere

Figure 1: **Multi-Traversal Gaussian Splatting (MTGS)** could reconstruct high-fidelity driving scenes from multi-traversal data. *All* images are *rendered* from a MTGS model of the *same* road block. **(a)** This approach preferably handles variations in lighting and shadows, rendering views conditioned on the traversal index (`Trv #`). **(b)** The extrapolation quality of MTGS is showcased. It maintains high visual quality, even with lateral shifts of 8 meters (*i.e.*, two lanes). For clarity, we mark a fixed reference point across traversals with a red pin.

## Abstract

Multi-traversal data, commonly collected through daily commutes or by self-driving fleets, provides multiple viewpoints for scene reconstruction within a road block. This data offers significant potential for high-quality novel view synthesis, which is crucial for applications such as autonomous vehicle simulators. However, inherent challenges in multi-traversal data often result in suboptimal reconstruction quality, including variations in appearance and the presence of dynamic objects. To address these issues, we propose Multi-Traversal Gaussian Splatting (MTGS), a novel approach that reconstructs high-quality driving scenes from arbitrarily collected multi-traversal data by modeling a shared static geometry while separately handling dynamic elements and appearance variations. Our method employs a multi-traversal scene graph with a shared static node and traversal-specific dynamic nodes, complemented by color correction nodes with learnable spherical harmonics coefficient residuals. This approach enables high-fidelity novel view synthesis and provides flexibility to navigate any viewpoint. We conduct extensive experiments on a large-scale driving dataset, nuPlan, with multi-traversal data. Our results demonstrate that MTGS improves LPIPS by 23.5% and geometry accuracy by 46.3% compared to single-traversal baselines. Code will be publicly available.

# 1 INTRODUCTION

Building photorealistic simulators is crucial for developing safe and robust autonomous vehicles (AVs), which could be adopted to create digital twins for testing autonomous systems (Feng et al., 2023; Ljungbergh et al., 2024; Zhou et al., 2024a), or generate diverse data for training end-to-end planning algorithms (Hu et al., 2023; Chen et al., 2024; Gao et al., 2025). To achieve this goal, the fundamental requirement is to synthesize high-fidelity renderings from arbitrary viewpoints while accurately preserving dynamic elements of the driving environment.

Scene reconstruction from recorded sensor data of AVs has gained popularity in recent years for this purpose (Yang et al., 2023; Tonderski et al., 2024; Yan et al., 2024b; Chen et al., 2025). However, methods that rely on single traversal logs often suffer from poor view extrapolation quality (Hwang et al., 2024; Han et al., 2024). In contrast, multi-traversal data covers a wide range of views and thus holds potential for viewpoint generalization. Intuitively, reconstruction using multi-traversal data improves quality for viewpoints that deviate from the original sequence. This is because views distributed across multiple lanes provide richer geometric constraints, potentially enabling view interpolation across the entire drivable area.

Nonetheless, reconstructing a high-fidelity scene across multi-traversals is non-trivial. One characteristic of multi-traversal data is that it represents the same shared space, but the collection could span over a large time period. This indicates that effective interpolation across traversals applies to the spatial aspects of the scene only, corresponding to static 3D geometry, while temporal variations, such as scene dynamics and appearance, remain challenging to interpolate. In particular, the sunlight and weather can be mixed, resulting in different exposure, tone, white balance, and shadow. Furthermore, scene dynamics include both moving and parked vehicles, which are also time-variant. As a consequence, naive reconstruction approaches often struggle to model these inconsistencies, leading to blurred outputs or severe artifacts (Qin et al., 2024; Han et al., 2024; Fischer et al., 2024).

To this end, we propose Multi-Traversal Gaussian Splatting (MTGS), a novel approach designed to reconstruct dynamic multi-traversal scenes through 3D Gaussian Splatting and thus synthesize photorealistic extrapolated views, as depicted in fig. 1. Our approach leverages images from multi-traversal sequences to reconstruct a shared static geometry while separately modeling scene dynamics and appearance variations across different traversals. Specifically, we propose a multi-traversal scene graph that builds a shared static node, and dynamic nodes within sub-graphs corresponding to each traversal. This design enables dynamic objects across traversals to be modeled in parallel. In addition, a LiDAR-guided exposure alignment module is introduced to ensure consistent appearance within individual traversal images. We further integrate an appearance node into each traversal subgraph to capture appearance variations in the form of the residual spherical harmonics coefficient. Finally, multiple regularization losses are developed to enhance the geometric alignment between traversals.

To measure the view extrapolation performance of MTGS with prior works fairly, a dedicated benchmark on the public driving dataset, nuPlan (Karnchanachari et al., 2024), is constructed. We select road blocks with multi-traversal data distributed across multiple lanes and evaluate one isolated traversal with minimal spatial overlap with others. Compared to single-traversal reconstruction, our multi-traversal approach consistently improves performance as additional traversals are incorporated, achieving up to an 18.5% improvement on the pixel-level metric (SSIM), 23.5% on the feature-level metric (LPIPS), and 46.3% on the geometry-level metric (absolute depth relative error). Our method also outperforms state-of-the-art approaches across all evaluation metrics.

The contributions are summarized as follows:

- We propose MTGS with a novel multi-traversal scene graph, including a shared static node that represents background geometry, an appearance node to model various appearances, and a transient node to preserve dynamic information.

- MTGS enables high-fidelity reconstruction with extraordinary view extrapolation quality. We demonstrate that the MTGS achieves state-of-the-art performance in driving scene extrapolated view synthesis. It outperforms previous SOTA by 17.6% on SSIM, 42.4% on LPIPS and 35% on AbsRel.

## 2 RELATED WORK

**Driving scene reconstruction.** Recent approaches on driving scene reconstruction can be categorized into two paradigms: neural radiance fields (NeRF) (Mildenhall et al., 2021) and 3D Gaussian splatting (3DGS) (Kerbl et al., 2023) based methods. NeRF-based methods (Yang et al., 2023; Wu et al., 2023; Tonderski et al., 2024) have shown remarkable success in reconstructing static backgrounds and dynamic agents via neural feature grids. Recent advancements in 3DGS provide a more efficient solution. DrivingGaussian (Zhou et al., 2024c), HUGS (Zhou et al., 2024b), and Street Gaussians (Yan et al., 2024b) initialize dynamic objects using 3D bounding boxes and utilize the scene graph design that separates static backgrounds and dynamic objects to reconstruct driving scenes. Building upon this foundation, OmniRe (Chen et al., 2025) models cyclists and pedestrians using Deformable Gaussian (Jung et al., 2023) nodes and SMPL (Loper et al., 2015) nodes. SplatAD (Hess et al., 2025) explores LiDAR rasterization and solves the rolling shutter effect on both image and LiDAR to achieve better results. Yet, existing methods focus on the single traversal setting mainly, *i.e.*, training and evaluating the original video sequence in a view interpolation manner. This work extends the dynamic scene graph design to a multi-traversal setting and evaluates an extrapolated view of unseen traversal.

**Novel view synthesis in autonomous driving.** It emphasizes the extrapolation ability in reconstruction models. This topic follows two technical paradigms primarily: regularization-guided and generative-prior-guided. Among regularization-based methods, AutoSplat (Khan et al., 2024) introduces planar assumptions on the geometry of the road and sky while exploiting the symmetry of foreground objects to reconstruct unseen parts. Vid2Sim (Xie et al., 2025) enforces patch-normalized depth consistency and adjacent pixel normal vector alignment. Recent research (Yu et al., 2025; Hwang et al., 2024; Yan et al., 2025) generates novel views with diffusion models conditioned on different features, *e.g.*, images, depth, or LiDAR, to supplement training 3DGS. FreeSim (Fan et al., 2025) extends the coverage limit of LiDAR and adopts a hybrid generative reconstruction method to add generated views to the reconstruction process progressively. StreetUnveiler (Xu et al., 2025) removes parking cars, reconstructs occluded background with an inpainting diffusion model, and designs a near-to-far sampling strategy to improve temporal consistency. While these methods achieve photorealistic synthesis, they exhibit prohibitive computational costs and are limited by the quality of generation models. They also fall short of flexibility when inpainting unseen or occluded parts. Our work addresses these gaps through multi-traversal images collected from the real world, and utilizes regularization to achieve better geometry consistency.

**Appearance modeling.** It has been a long-standing challenge in neural scene reconstruction. NeRF in the wild (Martin-Brualla et al., 2021) pioneered appearance modeling for unstructured photo collections by presenting learnable per-image appearance embeddings. Block-NeRF (Tancik et al., 2022) utilizes camera exposure parameters to optimize per-image appearance embeddings, enabling city-scale reconstruction. Recent works (Zhang et al., 2024; Lin et al., 2025; Xu et al., 2024) in 3DGS explore appearance modeling similarly. For instance, Kulhanek et al. (2024) propose to combine per-Gaussian and per-image appearance embeddings to model appearance variation. These methods aim to solve per-camera appearance alignment with large overlapped regions, or with additional information input. We address the challenge of appearance modeling in multi-traversal AV sensor datasets, characterized by unbounded, non-object-centric, and dynamic scenes. Our approach leverages the inherent appearance consistency within individual traversals and the variations observed across multi-traversals to achieve improved appearance modeling.

**Multi-traversal street reconstruction.** It builds scalable and robust 3D environmental representations for autonomous driving. Qin et al. (2024) employ semantic segmentation to mask transient objects out and learn per-traversal appearance embeddings. 3DGM (Li et al., 2024) proposes a self-supervised scene decomposition and mapping framework that leverages repeated traversals and pre-trained vision features to identify static backgrounds. The EUVS benchmark (Han et al., 2024) is designed to evaluate view extrapolation quality using multi-traversal data. It also includes a baseline that trains on multi-traversal data confined to a single lane. Existing methods tend to produce blurred synthesized outputs, primarily due to their simplistic modeling of scene dynamics and appearance variations. They also filter out all dynamic objects during reconstruction, while we contend that preserving them is essential for achieving a comprehensive reconstruction and enabling downstream applications.

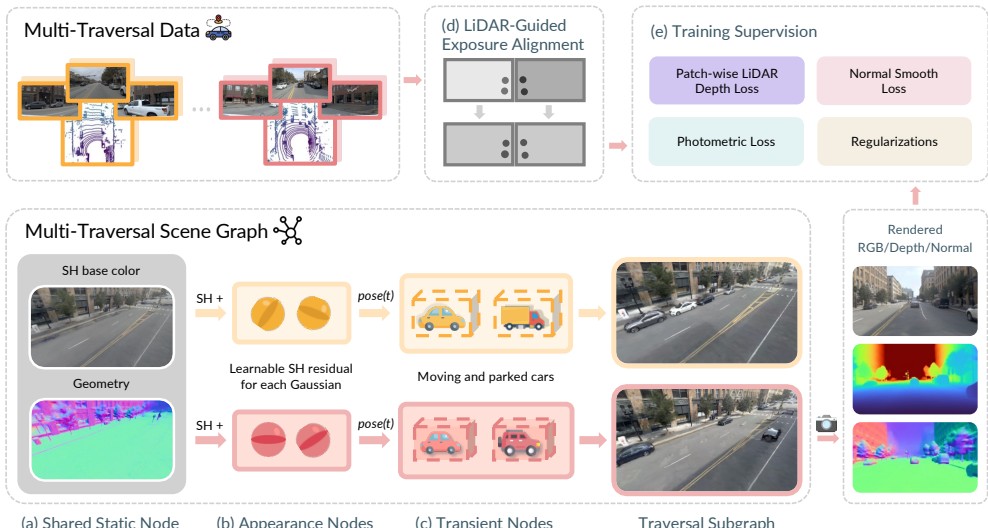

Figure 2: **Overview.** MTGS reconstructs a scene graph from multi-traversal sensor sequences. The scene graph consists of three types of nodes. **(a)** The rendering of a traversal subgraph starts with a shared static node, representing the time-invariant part of the scene. **(b)** This is followed by an appearance node that applies traversal-specific appearance effects, such as lighting and shadows. **(c)** Finally, transient nodes are placed in the background. **(d)** We align exposure using the overlapping LiDAR point cloud to ensure lighting consistency within the subgraph. **(e)** Photometric loss and multiple geometric losses are applied to bootstrap the reconstruction fidelity.

## 3 MULTI-TRAVERSAL GAUSSIAN SPLATTING

The overall pipeline of Multi-Traversal Gaussian Splatting (MTGS) is depicted in Fig. 2. MTGS reconstructs a Multi-Traversal Scene Graph from a set of multi-traversal data, enabling the generation of high-fidelity images. In this section, we first describe the problem formulation of multi-traversal reconstruction tasks. Then, we introduce the design of the Multi-Traversal Scene Graph. Next, we describe our approach for tuning appearances across multiple traversals. Finally, we detail the geometric regularization techniques employed in MTGS and training objectives.

### 3.1 PROBLEM FORMULATION

**Inputs.** The inputs for the task are videos captured in the same block but in different times. In other words, images $\mathcal{I} = \left\{ \mathbf{I}_{t,T} \in \mathbb{R}^{w \times h \times 3} \mid t = 0, \cdots, t_T; T = 1, \cdots, T_{\texttt{all}} \right\}$ are given with corresponding camera poses $\boldsymbol{\xi}_{t,T} = \left\{ \mathbf{W}_{t,T}, \mathbf{K}_{t,T} \right\}$, where $t$ represents time and $T$ represents traversals. Colored LiDAR point clouds $\mathbf{P}_{t,T} \in \mathbb{R}^{K \times 6}$ are also provided for initialization and sparse depth priors. Additionally, 3D bounding boxes of transient objects are included for scene graph initialization.

**Assumptions.** We assume that across multiple traversals in the same road block, the background remains largely consistent, *i.e.*, sharing the same geometry despite variations in appearance. Meanwhile, foregrounds, such as moving vehicles and parked cars along the streets, are traversal-variant.

**Outputs.** The output of the problem is a scene representation $f$ that can render the result $f(\boldsymbol{\xi}, t, T) \in \mathbb{R}^{w \times h \times C}$ captured by camera $\boldsymbol{\xi}$ at time $t$ in traversal $T$, where $0 \leq t \leq t_T$. $C$ represents the number of expected channels, including RGB, depth, *etc*. Note that the representation should also be able to render results for unseen traversals.

**Targets.** The target is to optimize $f$ so that its rendered results $f(\boldsymbol{\xi}, t, T)$ are as close as the ground truths of RGB, depth, *etc*., captured by camera $\boldsymbol{\xi}$ at time $t$ in traversal $T$.

## 3.2 Multi-Traversal Scene Graph

In multi-traversal settings, the state of the scene is determined by time $t$ and traversal $T$. To model transient objects and appearance changes, we represent the whole scene as a multi-traversal scene graph built upon 3DGS (Kerbl et al., 2023), containing three types of nodes, one shared static node for backgrounds, $\mathcal{G}^{\texttt{static}}$, multiple appearance nodes for backgrounds, $\mathcal{G}^{\texttt{appr}}_T$ for traversal $T$, and multiple transient nodes that exist in exactly one traversal, $\mathcal{G}^{\texttt{tsnt}}_{T,k}$ for the $k$-th node in traversal $T$. The scene in traversal $T$ is thus a subgraph composed of the shared static node $\mathcal{G}^{\texttt{static}}$, one appearance node $\mathcal{G}^{\texttt{appr}}_T$ and all transient nodes in the current traversal.

**Static node and appearance node.** For $G_i$ in the static backgrounds, $\mathcal{G}^{\texttt{static}}$ provides traversal-invariant and time-invariant properties $\{\mathbf{x}_i, \mathbf{q}_i, \mathbf{s}_i, \alpha_i, \boldsymbol{\beta}^{\texttt{base}}_i\}$ while the appearance node $\mathcal{G}^{\texttt{appr}}_T$ provides traversal-wise color residuals in traversal $T$, $\{\boldsymbol{\beta}^{\texttt{residual}}_{i,T}\}$. Here, for $G_i$ in traversal $T$,

$$\boldsymbol{\beta}_{i,0,0} = \boldsymbol{\beta}^{\texttt{base}}_i, \text{ and } \{\boldsymbol{\beta}_{i,l,m}\}^{-l \leq m \leq l}_{1 \leq l \leq l_{\max}} = \boldsymbol{\beta}^{\texttt{residual}}_{i,T}, \tag{1}$$

where $l$ and $m$ are the degree and order of the spherical harmonics $\{Y_{l,m}\}$. As for unseen traversals, the appearance node of its nearest training traversal is used. With such designs, only the coefficient of the rotation-invariant spherical harmonic (SH) $Y_{0,0}$ is shared across traversals. This encourages each Gaussian to encode the appearance component that remains consistent among all traversals in $Y_{0,0}$, effectively capturing its traversal-invariant color and enabling stable background geometry alignment. In contrast, appearance variations that differ across traversals, *e.g.*, lighting changes, reflections, or global color tone shifts, are absorbed by the traversal-specific residual coefficients $\boldsymbol{\beta}^{\texttt{residual}}_{i,T}$. These residuals are modeled using the higher-order, view-dependent SHs $\{Y_{l,m}\}_{1 \leq l \leq l_{\max}, -l \leq m \leq l}$, allowing it to flexibly represent traversal-dependent lighting and other view-dependent effects.

In contrast, if some coefficients in $\boldsymbol{\beta}^{\texttt{residual}}_{i,T}$ are shared, changes caused by various traversals would be mistaken for those caused by various views. When no SH coefficients are shared, the geometry of backgrounds is not aligned, leading to undesired background deviations across traversals.

**Transient node.** For Gaussians in $\mathcal{G}^{\texttt{tsnt}}_{T,k}$, $\mathbf{x}_i$ and $\mathbf{q}_i$ are defined in local coordinates of the node, and can be transformed into world coordinates by

$$\mathbf{x}^{\texttt{world}}_i(t) = \mathbf{R}_{T,k}(t)\mathbf{x}_i + \mathbf{T}_{T,k}(t), \tag{2}$$

$$\mathbf{q}^{\texttt{world}}_i(t) = \texttt{RotToQuat}(\mathbf{R}_{T,k}(t))\mathbf{q}_i, \tag{3}$$

where $\mathbf{R}_{T,k}(t)$ and $\mathbf{T}_{T,k}(t)$ are the rotation matrix and translation of the pose transform of the transient node over time, while $\texttt{RotToQuat}(\cdot)$ converts a rotation matrix into its corresponding quaternion.

To prevent transient nodes from using floaters to overfit the backgrounds, an out-of-box loss is also introduced as:

$$\mathcal{L}_{\texttt{oob}} = -\frac{1}{|\mathcal{G}^{\texttt{oob}}_{T,k}|} \sum_{G_i \in \mathcal{G}^{\texttt{oob}}_{k,T}} \log\left(1 - \alpha_i\right), \tag{4}$$

where $\mathcal{G}^{\texttt{oob}}_{T,k}$ is the set of Gaussians whose distance from the origin in the local coordinates is larger than $\frac{1}{2}S^{\texttt{tsnt}}_{T,k} + \theta_{\texttt{tol}}$. Here, $\theta_{\texttt{tol}}$ acts as a tolerance threshold so that the shadow of the foreground is contained in the transient node.

**Initialization.** We initialize the scene graph structure with automatically labeled 3D bounding bounding boxes from the dataset (Karnchanachari et al., 2024). From 3D boxes, we get transient nodes along with their sizes and transformations of poses over time. Gaussian points are initialized from aggregated LiDAR point clouds, with background and transient objects separated. Additionally, we employ point triangulation to initialize far-away Gaussians and randomly sample points on a semisphere to initialize Gaussians representing the sky.

**Scene decomposition.** We observe that reconstructing transient objects with such subgraph design, rather than simply masking them out, leads to better static reconstruction by preventing the background from overfitting on shadows of transients. Moreover, by this design, all transient objects, not just dynamic ones, can be decomposed from the background and are clearly reconstructed. For example, parked vehicles can be decoupled, as shown in Fig. 2.

## 3.3 Appearance Modeling

In the multi-traversal setting, appearance modeling is two-fold, the alignment within a traversal and the appearance tuning across multiple traversals. For appearance tuning, we propose appearance nodes in the scene graph to adjust the appearance of backgrounds (See section 3.2). For alignment within traversals, we introduce LiDAR-guided exposure alignment and learnable per-camera affine transforms.

**LiDAR-guided exposure alignment.** Images might vary in exposure due to various lighting. To normalize exposure across multi-camera images taken simultaneously, we use colored LiDAR points as photometric anchors. At each time $t$, LiDAR points are projected into all camera views to identify overlapping regions. Pixel intensities corresponding to the same 3D point are extracted and converted to HSV color space. Brightness ratios (V-channel) between overlapping camera pairs are computed, and exposures are iteratively adjusted to bring all views toward a common mean exposure level. This ensures a consistent appearance across views for reliable cross-camera fusion.

**Learnable per-camera affine transforms.** To enhance consistency between images taken at different times within one traversal, a per-camera affine (Martin-Brualla et al., 2021) transform $\texttt{Aff}(\cdot)$ is attached to refine the color tone, brightness, contrast, and exposure of image $\mathbf{I}_{\text{idx}} \in \mathcal{I}$ by:

$$\texttt{Aff}(\mathbf{I}) = \mathbf{W}_{\text{idx}}\mathbf{I} + \mathbf{b}_{\text{idx}}. \tag{5}$$

Note that learnable $\mathbf{W}_{\text{idx}} \in \mathbb{R}^{3 \times 3}$ and $\mathbf{b}_{\text{idx}} \in \mathbb{R}^3$ are image-wise, *i.e.*, different parameters for different images.

## 3.4 Regularization and training

To achieve high-quality 3D reconstruction and ensure consistency in geometry, we introduce two types of regularization: depth regularization and normal regularization.

**Patch-wise LiDAR depth loss.** The LiDAR depth loss contains an inverse L1 loss and a patch-wise normalized cross-correlation loss. We project sparse LiDAR points into the image plane to obtain sparse LiDAR depth as ground truth. The loss function for this regularization is defined as:

$$\mathcal{L}_{\text{depth}} = \left| \frac{1}{d_{\text{pred}}} - \frac{1}{d_{\text{LiDAR}}} \right|, \tag{6}$$

where $d_{\text{pred}}$ is the predicted depth and $d_{\text{LiDAR}}$ is the corresponding LiDAR depth.

However, depth from sparse LiDAR points can lead to local overfitting and discontinuity. To address this, we leverage a pre-trained dense depth estimator (Piccinelli et al., 2024) and enforce a patch-based normalized cross-correlation (NCC) depth regularization (Xie et al., 2025). NCC evaluates the similarity between scale-ambiguous pseudo depth and rendered depth patches, ensuring local consistency in depth rendering:

$$\mathcal{L}_{\text{ncc}} = 1 - \frac{1}{|\Omega|} \sum_{p \in \Omega} \sum_{s=1}^{S^2} \frac{\overline{D}_{p,s} D_{p,s}}{\overline{\sigma}_p \sigma_p}, \tag{7}$$

where $\Omega$ is the patches set of depth map with size $s \times s$ and stride $k$. $D_{p,s}$ and $\sigma_p$ represent a depth patch's mean-centered values and standard deviations, respectively.

**Normal smooth loss.** To define the normal of a Gaussian, we first note that a Gaussian itself does not inherently possess a normal direction. However, we can derive a geometric normal based on its ellipsoidal shape. Specifically, the normal is defined as the direction of the smallest scaling axis of the Gaussian, which corresponds to its shortest axis in 3D space. Inspired by DN-Splatter (Turkulainen et al., 2025), for a Gaussian described by a rotation matrix $\mathbf{R} \in \mathbb{R}^{3 \times 3}$ and a scaling vector $\mathbf{s}_i = [s_{i,0}, s_{i,1}, s_{i,2}] \in \mathbb{R}^3$, the normal is computed mathematically as:

$$\hat{\mathbf{n}}_{i,p} = \mathbf{R} \cdot \texttt{OneHot}\big(\text{argmin}(s_{i,0}, s_{i,1}, s_{i,2})\big), \tag{8}$$

where $\texttt{OneHot}(\cdot) \in \mathbb{R}^3$ returns a unit vector with all zeros except at the position of minimum scaling. To generate per-pixel normal estimates, the corrected normals of 3D Gaussians are first transformed into camera space using the current camera transformation matrix. A per-pixel normal $\hat{N}$ is computed via alpha compositing.

Figure 3: **Visual comparison.** Compared to OmniRE (Chen et al., 2025) and 3DGS (Kerbl et al., 2023), MTGS produces images in higher fidelity, effectively handles appearance variations, and robustly extrapolates to novel views (top). Notably, our transient node accurately captures moving shadows (red box).

The pseudo normal $N$ is estimated from the pseudo-depth map, as in 2DGS (Huang et al., 2024). To deal with noise in the pseudo normal, we introduce a total variation (TV) loss on the renderer normal. The normal regularization loss is:

$$\mathcal{L}_{\text{normal}} = |\hat{N} - N| + \mathcal{L}_{\text{TV}}(\hat{N}). \tag{9}$$

To obtain a stable Gaussian normal, we add a Gaussian flatten regularization loss to regularize the ratio of the other two axes not exceeding $r$ and minimize the minimum scale axis:

$$\mathcal{L}_{\text{flatten}} = \sum_i \texttt{max}\left\{\frac{\text{max}(\mathbf{s}_i)}{\text{median}(\mathbf{s}_i)}, r\right\} - r + \min(\mathbf{s}_i). \tag{10}$$

In the end, all components of MTGS are optimized jointly using the overall training loss:

$$\begin{aligned}\mathcal{L} = &\lambda_r \mathcal{L}_1 + (1 - \lambda_r)\mathcal{L}_{\text{SSIM}} + \lambda_{\text{depth}}\mathcal{L}_{\text{depth}} + \lambda_{\text{ncc}}\mathcal{L}_{\text{ncc}} \\ &+ \lambda_{\text{normal}}\mathcal{L}_{\text{normal}} + \lambda_{\text{flatten}}\mathcal{L}_{\text{flatten}} + \lambda_{\text{oob}}\mathcal{L}_{\text{oob}},\end{aligned} \tag{11}$$

where $\mathcal{L}_1$ and $\mathcal{L}_{\text{SSIM}}$ are photometric losses between ground truth images and renderer images, $\lambda_r$, $\lambda_{\text{depth}}$, $\lambda_{\text{ncc}}$, $\lambda_{\text{normal}}$, $\lambda_{\text{flatten}}$, and $\lambda_{\text{oob}}$ are hyper-parameters.

## 4 EXPERIMENT

### 4.1 SETUP AND PROTOCOLS

**Dataset.** The experiments are conducted on dedicated multi-traversal data extracted from nuPlan (Karnchanachari et al., 2024). This large-scale driving dataset comprises over 100 hours of data, featuring eight surrounding-view images and point clouds merged from 5 LiDAR sensors. We use all eight views and LiDAR at 10 Hz, with the resolution of $960 \times 540$ for images across training and evaluation. We select six road blocks with multi-traversal data distributed across multiple lanes and pick one isolated traversal with minimal spatial overlap with others for novel-view evaluation.

**Implementation details.** Our method is implemented upon open-source repositories, nerfstudio and gsplat (Tancik et al., 2023; Ye et al., 2025). For simplicity, non-rigid dynamics are ignored in reconstruction. We select three baselines, 3DGS (Kerbl et al., 2023), Street Gaussians (Yan et al., 2024b), and OmniRe (Chen et al., 2025). The 3DGS baseline is implemented in gsplat, while other baselines are adapted from OmniRe's codebase. By default, we train all methods with 30k steps using Adam optimizers. For details, please refer to the supplementary.

**Metrics.** We compute metrics on three aspects of both training traversal and novel-view traversal.

Table 1: **Comparison with SOTA.** 'ST' denotes single-traversal reconstruction. 'MT' stands for multi-traversal reconstruction. For MT, results on training traversals are averaged and cannot be compared with those in ST directly. The novel-view traversal is identical between ST and MT. ∗: affine-aligned PSNR. †: adapted with multi-traversal transient nodes. First, second, third.

|   | Method | Training Traversal | | | | Novel-View Traversal | | | | | |
|---|---|---|---|---|---|---|---|---|---|---|---|
|   |   | PSNR ↑ | SSIM ↑ | LPIPS ↓ | AbsRel ↓ | PSNR* ↑ | SSIM ↑ | LPIPS ↓ | Feat. Sim. ↑ | AbsRel ↓ | Delta1 ↑ |
| ST | 3DGS | 25.40 | 0.775 | 0.299 | 0.256 | 19.15 | 0.570 | 0.414 | 0.514 | 0.285 | 0.437 |
|    | StreetGS | 23.32 | 0.852 | 0.304 | 0.080 | 17.39 | 0.473 | 0.479 | 0.558 | 0.157 | 0.815 |
|    | OmniRe | 23.64 | 0.865 | 0.283 | 0.081 | 17.34 | 0.466 | 0.474 | 0.560 | 0.162 | 0.805 |
|    | **Ours** | 29.43 | 0.879 | 0.150 | 0.094 | 20.11 | 0.575 | 0.313 | 0.614 | 0.145 | 0.879 |
| MT | 3DGS | 22.04 | 0.705 | 0.390 | 0.332 | 20.53 | 0.614 | 0.388 | 0.557 | 0.347 | 0.312 |
|    | StreetGS† | 20.57 | 0.736 | 0.447 | 0.097 | 18.18 | 0.527 | 0.488 | 0.577 | 0.148 | 0.826 |
|    | OmniRe† | 20.91 | 0.755 | 0.409 | 0.092 | 18.36 | 0.527 | 0.460 | 0.594 | 0.136 | 0.859 |
|    | **Ours** | 28.04 | 0.848 | 0.192 | 0.094 | 21.65 | 0.628 | 0.265 | 0.670 | 0.089 | 0.904 |
|    | **Ours (60k)** | 28.73 | 0.865 | 0.169 | 0.094 | 21.58 | 0.620 | 0.254 | 0.676 | 0.091 | 0.902 |

- Pixel-level metrics. We use peak signal-to-noise ratio (PSNR), structural similarity index measure (SSIM) (Wang et al., 2003), and affine-aligned PSNR (Barron et al., 2022) for novel-view traversals.

- Feature-level metrics. We employ learned perceptual image patch similarity (LPIPS) (Zhang et al., 2018) and DINOv2 (Oquab et al., 2023) feature cosine similarity (Feat. Sim.), which matters more to the downstream visual models (Lindström et al., 2024).

- Geometry-level metrics. We evaluate geometry accuracy with depth-related metrics, including the absolute relative error and $\delta_{1.25}$ (delta 1), between the rendered depth and projected LiDAR depth within an 80-meter range.

## 4.2 MAIN RESULTS

We show results in both single-traversal (ST) and multi-traversal (MT) settings in Tab. 1. In ST tests, our method outperforms others in image reconstruction, likely due to its effective inner-traversal appearance modeling. It also achieves the highest quality in novel-view synthesis across all metrics, especially at feature and geometry levels.

In the MT setting, MTGS reconstructs a consistent scene across multiple traversals. Although OmniRe obtains good results on training traversals regarding SSIM and AbsRel, its severe overfitting leads to poor performance on novel-view traversals. In contrast, MTGS consistently delivers the best novel-view synthesis performance. Notably, with additional training iterations (60k), the feature-level metrics further improve, while the geometry metrics tend to converge. A qualitative comparison is shown in Fig. 3. Baselines produce blurry, artifact-prone images, whereas our method delivers clear, crisp results under both training-view and novel-view.

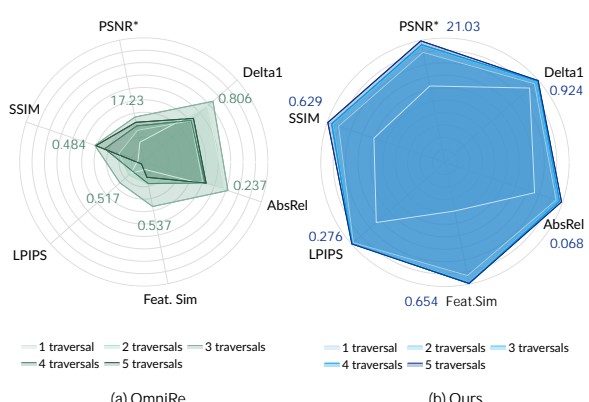

Figure 4: **Novel-view performance when trained with more traversals.** Extra traversals yield limited gains in baseline, while our design leverages them.

## 4.3 ABLATION STUDY

**Number of traversals.** We conduct experiments on three road blocks with six traversals. Traversals in these blocks are not occluded by any buildings or obstructions on-road to ensure that the performance gain of multi-traversal is not simply from seeing the unseen part. As shown in Fig. 4, our method enhances overall rendering quality on novel-view traversals as more traversals are incorporated. In contrast, OmniRe fails to maintain consistent geometry with the increased data. This demonstrates that MTGS effectively manages the appearance and dynamic variation across multiple

Table 2: **Ablation on appearance modeling.** The full model in ID 5 validates the effectiveness of our designs, as well as outperforming existing methods in handling the appearance variations (Kulhanek et al., 2024). 'CamAFF' refers to per-camera affine, 'LEA' denotes LiDAR exposure alignment, and 'Appr.Node' represents the appearance node. First , second , third .

| ID | Module | | | Training Traversal | | | | Novel-View Traversal | | | | | |
|---|---|---|---|---|---|---|---|---|---|---|---|---|---|
| | CamAFF | LEA | Appr. Node | PSNR ↑ | SSIM ↑ | LPIPS ↓ | AbsRel ↓ | PSNR* ↑ | SSIM ↑ | LPIPS ↓ | Feat. Sim. ↑ | AbsRel ↓ | Delta1 ↑ |
| 0 | | | | 23.46 | 0.784 | 0.249 | 0.105 | 19.66 | 0.582 | 0.308 | 0.612 | 0.094 | 0.901 |
| 1 | ✓ | | | 24.95 | 0.799 | 0.229 | 0.090 | 20.05 | 0.589 | 0.293 | 0.628 | 0.082 | 0.912 |
| 2 | ✓ | ✓ | | 26.16 | 0.818 | 0.219 | 0.086 | 20.85 | 0.615 | 0.281 | 0.633 | 0.078 | 0.914 |
| 3 | ✓ | | ✓ | 27.08 | 0.838 | 0.199 | 0.087 | 20.02 | 0.586 | 0.288 | 0.638 | 0.078 | 0.914 |
| 4 | | ✓ | ✓ | 27.59 | 0.853 | 0.190 | 0.084 | 20.79 | 0.612 | 0.274 | 0.641 | 0.077 | 0.914 |
| 5 | ✓ | ✓ | ✓ | 28.51 | 0.859 | 0.179 | 0.085 | 20.83 | 0.611 | 0.271 | 0.646 | 0.078 | 0.914 |
| 6 | WildGaussians | | | 25.20 | 0.805 | 0.229 | 0.096 | 19.88 | 0.577 | 0.300 | 0.618 | 0.087 | 0.909 |

traversals, resulting in a more accurate reconstruction of the shared static node. Full results are in the Supplement.

**Multi-traversal appearance modeling.** In Tab. 2, we demonstrate the effectiveness of our proposed appearance modeling designs by selecting a challenging subset of four road blocks, each containing three training traversals with various appearances. Removing modules from the final design (ID 2-4, compared to ID 5) leads to a performance drop, while incrementally adding modules to the baseline (ID 0-2, and 5) yields significant gains. Removing the appearance node (ID 2) significantly degrades training traversal reconstruction. On novel-view traversal, pixel-wise metrics drop slightly with the appearance node due to the applied nearest-traversal appearance, but feature-level performance improves notably, reflecting more realistic results. These findings validate that our strategy effectively captures and reconstructs the diverse appearances across multiple traversals, thereby enhancing both image reconstruction and novel-view synthesis. We also compare our approach with a state-of-the-art Gaussian-based in-the-wild method, WildGaussians (Kulhanek et al., 2024). We re-implement its per-camera and per-gaussian appearance embeddings within our pipeline. The results reveal that modeling per-camera appearance in a multi-traversal setting is insufficient, as the limited overlapping regions between cameras complicate the optimization process.

**Modular design.** We further evaluate additional design choices in MTGS. As shown in Tab. 3, removing the transient node (ID 0) degrades geometry accuracy, likely due to overfitting on the shadows cast by dynamic objects. These results demonstrate that preserving and modeling dynamic information can help multi-traversal reconstruction performance. Removing the normal smooth loss (ID 1) adversely affects the feature-level metrics, while removing the depth loss (ID 2) significantly harms learning the geometry.

Table 3: **Ablation on modular designs.** Nodes in the scene graph and regularization losses are all crucial for the final performance. 'tsnt.' stands for transient. First , second .

| ID | Exp. name | Novel-View Traversal | | | | | |
|---|---|---|---|---|---|---|---|
| | | PSNR* ↑ | SSIM ↑ | LPIPS ↓ | Feat. Sim. ↑ | AbsRel ↓ | Delta1 ↑ |
| 0 | w/o tsnt. node | 20.62 | 0.607 | 0.276 | 0.642 | 0.086 | 0.899 |
| 1 | w/o normal loss | 20.82 | 0.614 | 0.275 | 0.643 | 0.076 | 0.914 |
| 2 | w/o depth loss | 20.83 | 0.607 | 0.264 | 0.644 | 0.891 | 0.613 |
| 3 | Ours (Full) | 20.83 | 0.611 | 0.271 | 0.646 | 0.078 | 0.914 |

## 5 CONCLUSION AND OUTLOOK

In this work, we propose Multi-Traversal Gaussian Splatting (MTGS), a novel method capable of reconstructing multi-traversal dynamic scenes with high fidelity. By introducing a novel Multi-Traversal Scene Graph, our approach effectively captures a shared static background while separately modeling dynamic objects and appearance variations across multiple traversals. Extensive evaluations demonstrate that MTGS achieves both high-quality image reconstruction and robust novel-view synthesis, outperforming existing state-of-the-art methods. With its potential to serve as a foundation for photorealistic autonomous driving simulators, MTGS promises to enhance the safety and reliability of autonomous vehicle testing and development.

**Limitation and future works.** MTGS does not explicitly address the extrapolated views of transient objects. Enhancing MTGS with the generative model is a promising direction. In the shared background, there might be floating artifacts in regions not observed during training traversals, such as the space below parked cars. Modeling and reconstruction of unlabeled transient objects and map

changes are left for future work. Future endeavors may include simultaneous camera and LiDAR simulation, *e.g.* modeling the appearance diversity of LiDAR intensity and drop rate.

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

# APPENDIX

In the Appendix, we provide implementation details of our data preparation pipeline, our method, and the baselines. Additional experiments and visualizations are also included. For transparency, we further include the **code** and **video visualization** in the supplementary materials.

## A  IMPLEMENTATION DETAILS

We provide key implementation details on datasets, models, and experiments in the supplementary material. To encourage and facilitate further research, we will openly release the whole suite of code and models.

### A.1  DATASET

We conduct experiments on customized data from the nuPlan dataset (Karnchanachari et al., 2024). We use all eight views and LiDAR at 10 Hz, with the resolution of $960 \times 540$ for images across training and evaluation.

**Handling of inaccurate pose alignment.** Since the localization across multiple traversals in nu-Plan is imprecise, we employ a LiDAR registration method (Vizzo et al., 2023) to align the multi-traversal poses accurately. The camera extrinsic is pre-calibrated but not perfectly synced with Li-DAR, causing a pose shift when the car moves. To fix this problem, we composite the motion to the camera extrinsic by interpolation. We further use a camera pose optimizer (Yan et al., 2024a) to handle this misalignment.

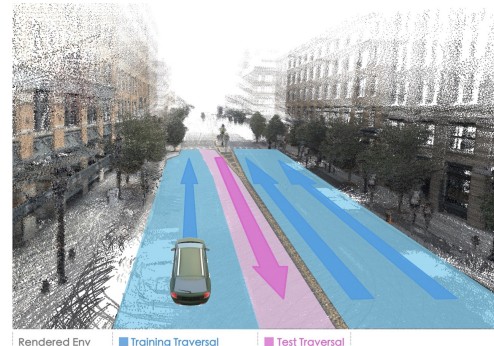

Figure A1: **Illustration of training and test traversals.** We select traversals distributed across multiple lanes and choose the isolated traversal with minimum overlaps.

**Handling of large image distortion.** We also note that the camera distortion in nuPlan is severe and could cause bad outputs as in the raw implementation of OmniRe (Chen et al., 2025). We undistort the images with OpenCV at optimal mode to preserve the field of view. To alleviate the inaccurate camera intrinsics in nuPlan, we employ several rounds of bundle adjustment of COLMAP (Schönberger and Frahm, 2016; Schönberger et al., 2016) to calibrate them.

**Use of pre-trained models.** The pseudo depth used during training is obtained from UniDepth (Piccinelli et al., 2024) with a ViT-L (Dosovitskiy et al., 2021) backbone. We input the undistorted images and the optimized focal length to UniDepth. Although it generates depth on a metric scale, the depth RMSE is still over 20 meters, which motivates us to apply the NCC loss in our model. To extract semantic masks, Mask2Former (Cheng et al., 2022) with a Swin-L (Liu et al., 2021) backbone trained on Cityscapes (Cordts et al., 2016) is adopted.

**Benchmark.** We list the road blocks used in experiments in Tab. A1. The traversals within road blocks are about 100 meters in length. The main comparison is based on all six traversals. The ablation on the number of traversals is based on traversals 0, 1, and 2. The rest of the ablations are based on traversals 0, 1, 2, and 5 with three training traversals. The principle of selecting traversals is shown in Fig. A1.

### A.2  MTGS

**Transient Node.** The initial poses for each transient node are derived from 3D bounding box annotations provided in the nuPlan dataset, which are generated by a pre-trained LiDAR 3D detector and tend to be inaccurate. Therefore, we treat these poses as learnable parameters, following the approach of Street Gaussians and OmniRe (Yan et al., 2024b; Chen et al., 2025), without applying a smoothness loss. The poses of static objects are kept the same across frames. An object with movements of less than 3 meters is considered static.

Table A1: **Details of selected road blocks.** The city name is from nuPlan (Karnchanachari et al., 2024). Coordinates are $x_{min}, y_{min}, x_{max}, y_{max}$ in UTM coordinate.

| ID | City | Road Block Coordinate | # Traversal |
|----|------|----------------------|-------------|
| 0 | us-ma-boston | 331220, 4690660, 331190, 4690710 | 6 |
| 1 | sg-one-north | 365000, 144000, 365100, 144080 | 3 |
| 2 | sg-one-north | 365530, 143960, 365630, 144060 | 4 |
| 3 | us-pa-pittsburgh-hazelwood | 587400, 4475700, 587480, 4475800 | 4 |
| 4 | us-pa-pittsburgh-hazelwood | 587640, 4475600, 587710, 4475660 | 6 |
| 5 | us-pa-pittsburgh-hazelwood | 587860, 4475510, 587910, 4475570 | 6 |

Table A2: **Details of training hyperparameters**

| Parameters | Initial LR | Final LR | Warm-up Steps |
|-----------|-----------|----------|---------------|
| means | 8e-4 | 8e-6 | 0 |
| static.features_dc | 1.25e-4 | 1.25e-4 | 0 |
| appearance.features_rest | 1.25e-4 | 1.25e-4 | 0 |
| transient.features_dc | 2.5e-3 | 2.5e-3 | 0 |
| transient.feature_rest | 1.25e-4 | 1.25e-4 | 0 |
| opacities | 5e-2 | 5e-2 | 0 |
| scales | 5e-3 | 5e-3 | 0 |
| quats | 1e-3 | 1e-3 | 0 |
| camera_pose_opt | 1e-4 | 5e-7 | 1500 |
| camera_affine | 1e-3 | 1e-4 | 5000 |
| ins_rotation | 1e-5 | 1e-6 | 0 |
| ins_translation | 5e-4 | 1e-4 | 0 |

**LiDAR-guided exposure alignment.** To ensure consistent appearance across all cameras, we perform exposure normalization for each frame using LiDAR points as photometric references. The following describes the practical implementation details used in our system.

After projecting LiDAR points into every camera, we determine which points are simultaneously visible in multiple views. These co-visible points give us natural correspondences across cameras without relying on image-based feature matching. For each pair of neighboring or spatially related cameras, we check how many LiDAR points fall inside both of their image bounds. If the shared set is sufficiently large, these points serve as photometric anchors. In rare frames where overlap is extremely small, we fall back to pre-defined boundary regions between the two cameras to approximate their shared field of view.

For the selected overlapping samples (either LiDAR-projected points or fallback regions), we read the corresponding pixel colors from both images. Each sampled color is converted to HSV space, and we focus specifically on the brightness (V-channel), which is more stable for cross-camera comparisons compared to raw RGB intensities. By comparing the average V-channel values from the two cameras, we estimate how much brighter or darker one camera is relative to the other.

The camera rig is treated as a connected graph, where each edge represents a pair of cameras with overlapping visibility. One camera is selected as a reference, and the relative brightness ratios computed from the overlap regions are used to propagate exposure adjustments throughout the graph. When multiple paths provide estimates for the same camera, their suggestions are averaged to increase stability and reduce noise. This produces an exposure scaling factor for every camera, ensuring that all views become consistent with respect to the reference camera. Finally, to avoid inadvertently shifting the overall exposure of the entire rig, we normalize all per-camera scaling factors to have a uniform global mean. This step preserves the intended brightness distribution of the scene while ensuring consistency between views. The resulting per-camera factors are applied to the V-channel of each image before converting back to RGB.

A potential concern is that view-dependent lighting effects (e.g., reflections or specular highlights) could be mistaken for exposure differences. In practice, our LiDAR-guided alignment is stable because the large overlapping regions between rig cameras provide many LiDAR-anchored samples,

allowing averaging to suppress outliers caused by such effects. This is especially effective since many anchors lie on ground surfaces, which exhibit minimal view-dependent variation. Additionally, learnable per-camera affine transforms serve as a fallback to handle any remaining inconsistencies.

**Optimization.** For the optimization process, we employ the Adam optimizer to train our model over 30,000 iterations. All the corresponding hyperparameters are explicitly outlined in Tab. A2. For Gaussian density control, we keep most of the hyperparameters as in the original 3DGS. Since we train the scene on the metric scale without normalization, we adjust the scale threshold of densify to 0.2 meters and the scale threshold of culling to 0.5 meters. To remove floaters, we set the gradient threshold of density to 0.001.

**Initialization.** We initialize a multi-traversal scene graph with metric scale points based on road block-centered coordinates. After aggregating all the LiDAR points, we first remove the statistical outlier to prevent floaters and then perform the voxel downsample with a size of 0.15 meters. We employ point triangulation to initialize far-away Gaussians. For the sky in the scene, we sample 100k points uniformly on a semisphere, with polar angles sampled from $[\frac{\pi}{4}, \frac{\pi}{2}]$ and a radius of two times for the farthest point of the scene.

**Losses.** Our model is optimized with $\lambda_r = 0.8$, $\lambda_{depth} = 0.5$, $\lambda_{ncc} = 0.1$, $\lambda_{normal} = 0.1$, $\lambda_{flatten} = 1.0$ and $\lambda_{obb} = 1.0$. In the NCC loss, patch size $s$ is set to 32, and $k$ is set to 16. In the Gaussian flatten loss, $r$ is set to 10 and is applied every 10 steps following (Xie et al., 2024).

## A.3 REPRODUCTION OF BASELINES

**3DGS (Kerbl et al., 2023).** We reproduce 3DGS based on gsplat (Ye et al., 2025). We set all the hyperparameters based on the original papers. The scene and the initialized point clouds are normalized with scale factor 5e-3, which corresponds to 200 meters scene extent.

**OmniRe (Chen et al., 2025).** We adopt its official implementation with default hyperparameters. We perform equivalent data preprocessing steps, including LiDAR registration, bundle adjustment, and distortion correction, which are consistent with our method. Notably, as we do not assess human body reconstruction, we omit the SMPL node component from OmniRe's pipeline and excluded pedestrians and bicycles during evaluation to ensure fairness.

**Street Gaussians (Yan et al., 2024b).** For Street Gaussians, we employ the implementation in the OmniRe repository and the default parameters while maintaining identical data processing protocols.

## B EXPERIMENTS

**Ablation on the extrinsic calibration.** As shown in Tab. A4, proper pose alignment significantly boosts both reconstruction and novel-view synthesis performance. Overfitting on inaccurate camera poses degrades view extrapolation. Since our pose alignment process is not fully optimized, improving multi-traversal localization represents a promising direction for enhanced reconstruction.

**Ablation on single-traversal setting.** MTGS outperms SOTA methods on single traversal setting in Tab. 1. This comes from multiple modules included in MTGS. Without these modules, the baseline model underperforms the SOTA method in the single traversal setting as in Tab. A3. The baseline includes LiDAR depth loss and learnable camera affinement to ensure fairness with OmniRe. Note that additional geometric constraints can prevent overfitting the training traversal and improve novel-view synthesis.

Table A3: **Ablation on single-traversal setting.**

| Method | Training Traversal | | Novel-View Traversal | | |
|---|---|---|---|---|---|
| | SSIM ↑ | AbsRel ↓ | SSIM ↑ | Feat. Sim. ↑ | AbsRel ↓ |
| OmniRe [3] | 0.865 | 0.081 | 0.466 | 0.560 | 0.162 |
| Baseline | 0.848 | 0.103 | 0.549 | 0.553 | 0.175 |
| + LiDAR Expo. Align. | 0.858 | 0.101 | 0.564 | 0.584 | 0.165 |
| + Camera Pose Opt. | 0.895 | 0.105 | 0.565 | 0.608 | 0.153 |
| + Geom. Cons. (MTGS) | 0.879 | 0.094 | 0.575 | 0.614 | 0.145 |

**Support for non-rigid reconstruction.** Non-rigid dynamic objects, such as pedestrian, are important roles on the road. However, nuPlan dataset has huge noise on the 3D bounding boxes of pedestrian, making the reconstruction problematic. For simplicity, we do not include non-right reconstruction in our experiments. Note that MTGS can seemlessly support non-rigid transient objects (*e.g.*

Table A4: **Ablation on extrinsic calibration.**

| | Module | | Training Traversal | | | | Novel-View Traversal | | | | | |
|---|---|---|---|---|---|---|---|---|---|---|---|---|
| ID | LiDAR Registration | CamOptim | PSNR ↑ | SSIM ↑ | LPIPS ↓ | AbsRel ↓ | PSNR* ↑ | SSIM ↑ | LPIPS ↓ | Feat. Sim.↑ | AbsRel ↓ | Delta1 ↑ |
| 0 | | | 25.45 | 0.776 | 0.298 | 0.131 | 19.43 | 0.572 | 0.374 | 0.519 | 0.177 | 0.700 |
| 1 | ✓ | | 27.70 | 0.837 | 0.199 | 0.082 | 20.79 | 0.613 | 0.281 | 0.639 | 0.078 | 0.914 |
| 2 | ✓ | ✓ | 28.51 | 0.859 | 0.179 | 0.085 | 20.83 | 0.611 | 0.271 | 0.646 | 0.078 | 0.914 |

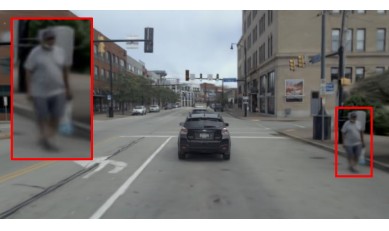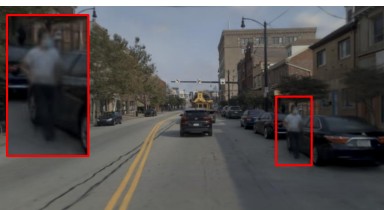

Figure A2: **MTGS reconstruction with deformable transient node in one road block.**

pedestrian) reconstuction by integrating techniques like deformable Gaussians (Jung et al., 2023) into the transient nodes, as in Figure A2.

**Computational requirement.** We report key computational metrics including rendering frames per second (FPS), optimization time, Gaussian count, and model size in Tab. A5. Thanks to the highly efficient implementation of our pipeline, MTGS

Table A5: **Comparision on computational requirement.**

| | FPS | Opt. Time (hour) | Gaussian Count (M) | Model Size (MiB) |
|---|---|---|---|---|
| OmniRe | 17.64 | 5.25 | 1.95 | 528.6 |
| MTGS | 144.3 | 2.40 | 2.48 | 1181 |

achieves higher FPS and shorter optimization time compared to OmniRe. Although the introduction of multi-traversal appearance nodes increases memory usage, the overall model size remains manageable.

**Reconstruction on big intersections.** Fig. A3 shows that MTGS can reconstruction big intersections with occlusions. We exclude such data from our evaluation to ensure that performance gains are not simply due to seeing the unseen regions.

**More visualization.** As shown in Fig. A4, we show more visualization on extrapolated views of our blocks. The visualization results of each block are arranged sequentially from left to right according to the temporal order of the traversal.

**Ablation on the number of traversals.** Results of all 7 metrics on both training and novel-view traversals are shown in Fig. A5.

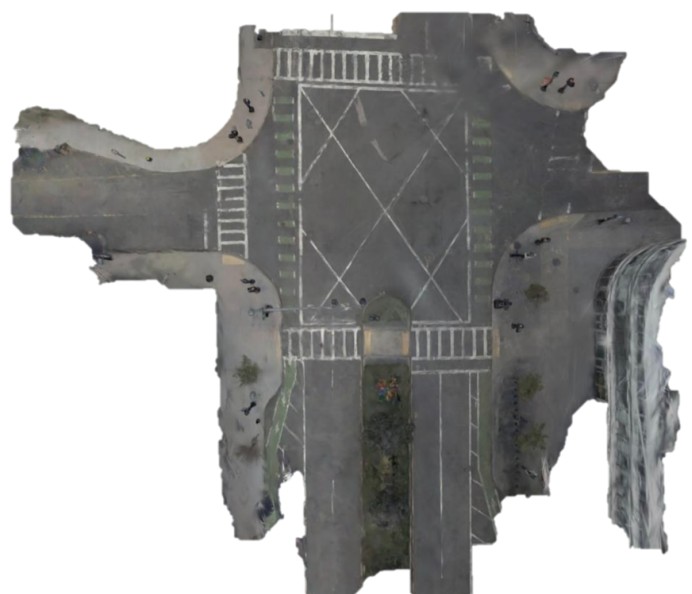

Figure A3: **An intersection with occluded areas.** MTGS can also reconstruct big intersections with occlusions, *e.g.*, buildings and road medians.

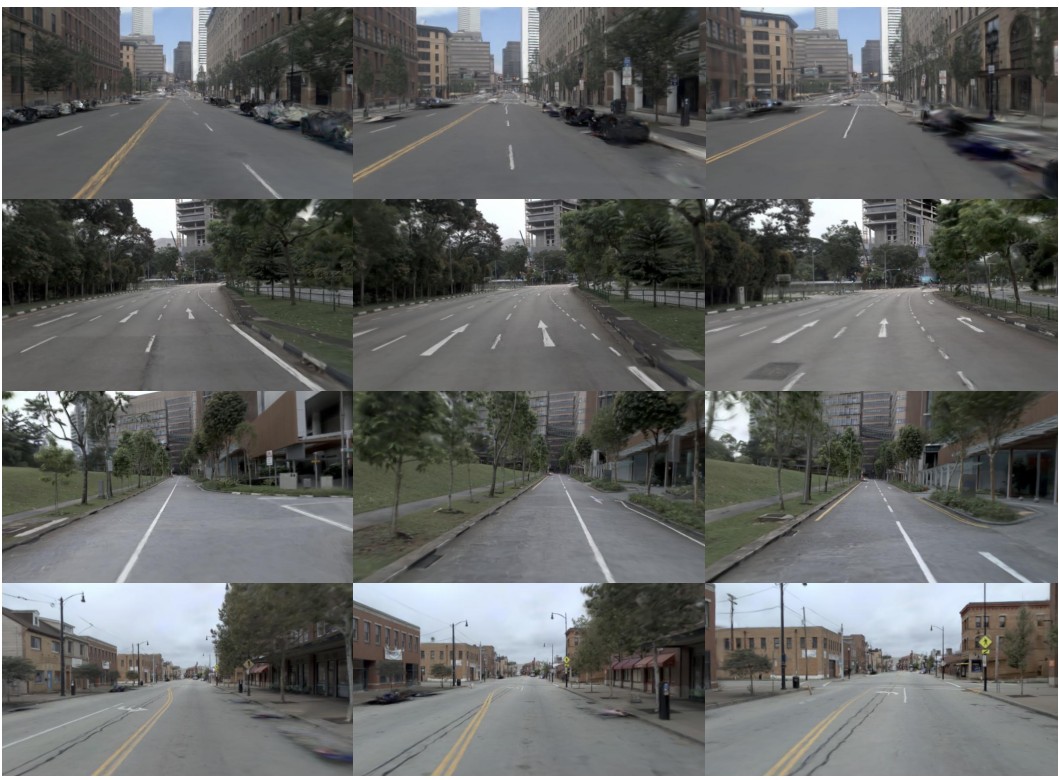

Figure A4: **More visualization on extrapolated views.** MTGS consistently generates high-quality view extrapolations. However, since all transient nodes are removed when rendering unseen trajectories, floating artifacts appear over car parking areas.

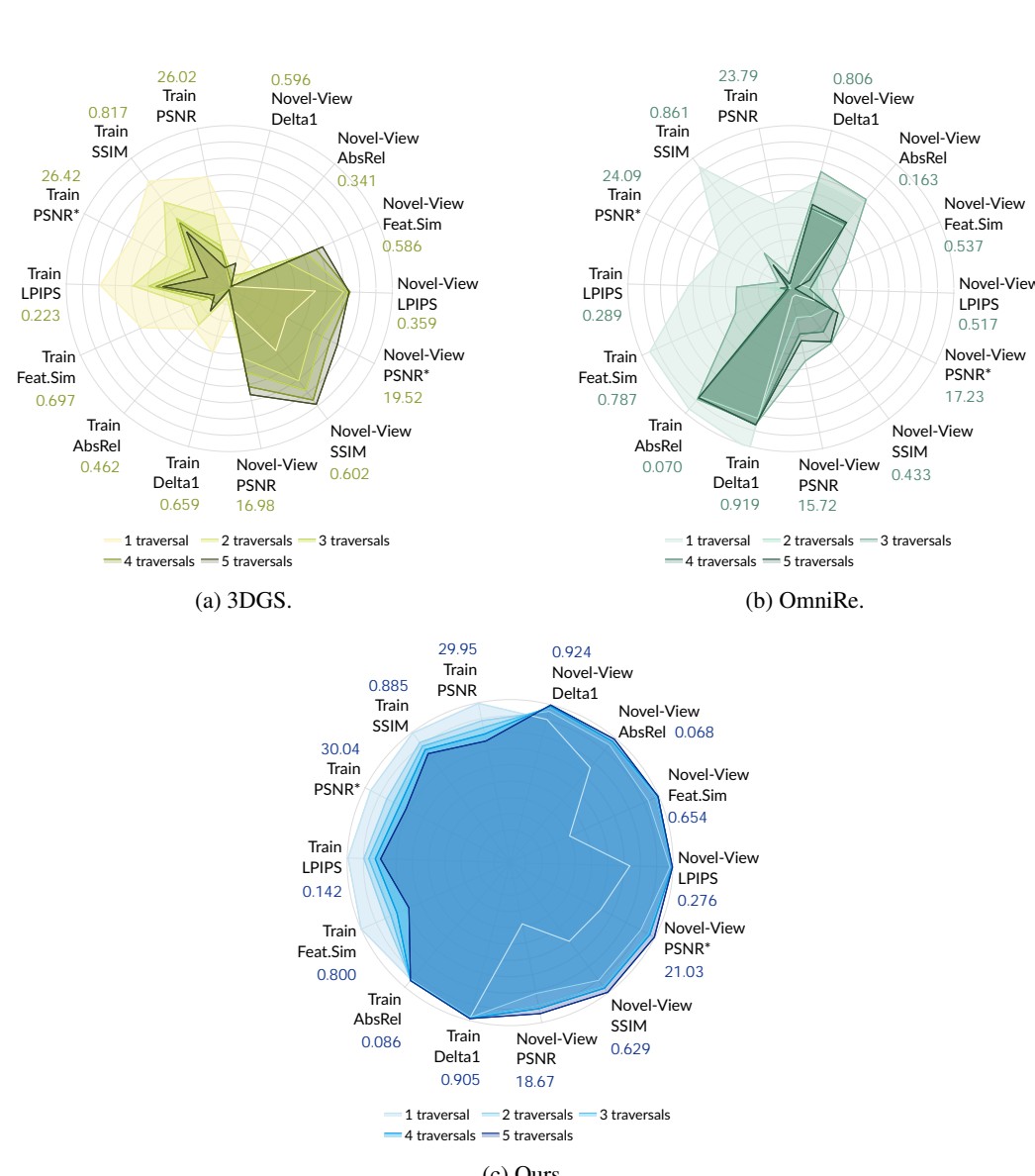

(a) 3DGS.

(b) OmniRe.

(c) Ours.

Figure A5: **Performances of three methods when trained on more traversals.** Note that outer rings represent better performance instead of larger scores. More traversals do not guarantee better performances for existing methods while our designs could continually benefit from more traversals used.

