# OpenReview forum: "MTGS: Multi-Traversal Gaussian Splatting"
_ICLR.cc/2026/Conference — Submitted to ICLR 2026_

### Official Review · Reviewer_Nsvs · 2025-10-23

**Soundness:** 3
**Presentation:** 3
**Contribution:** 2
**Rating:** 4
**Confidence:** 5

**Summary:**

Multi-traversal data enables multi-view road scene reconstruction and high-quality novel view synthesis (vital for autonomous vehicle simulators). However, it faces challenges like appearance variations and dynamic objects, leading to suboptimal reconstruction. We propose Multi-Traversal Gaussian Splatting (MTGS), which models shared static geometry while handling dynamic elements and appearance variations separately for high-fidelity novel view synthesis. Experiments on nuPlan show MTGS improves LPIPS by 23.5% and geometry accuracy by 46.3% over single-traversal baselines.

**Strengths:**

1. Interesting questions and clear definitions of the questions.
2. Clear expression of methods and presentation of algorithms.
3. Reasonable ablation experiments.

**Weaknesses:**

The paper has several areas that need improvement:
- **Lack of comparison with the latest methods**: The paper fails to compare with the latest novel view synthesis methods such as ReconDreamer, FreeVS, and Dist - 4D. By comparing with these methods, the performance, advantages, and disadvantages of the proposed method can be more comprehensively evaluated, providing more valuable reference for readers.
- **Absence of novel view synthesis visualization**: The paper does not provide novel view synthesis visualization. It is recommended to visualize translations of 1m, 2m, and 4m. This kind of visualization can more intuitively show the effect of the method in novel view synthesis, helping readers better understand the performance of the method.
- **Poor - quality images**: The image quality in the paper is poor and cannot be used in real - world scenarios, and there is a large gap compared with the novel view synthesis of ReconDreamer. High - quality images are crucial for demonstrating the effectiveness of the method, and poor - quality images may affect readers' understanding and evaluation of the method.
- **Poor - quality reconstruction results in the demo**: The reconstruction results in the demo seem to be of poor quality, and there is still a large gap compared with the demo of OmniRe. The demo is an important means to show the practical effect of the method. Poor - quality reconstruction results may make readers question the practical value of the method.

**Questions:**

The main problem lies in the above aspects. Additionally, the innovation of the thesis is limited.

---

> ### Author Response · Authors · 2025-11-25
> **Response to Reviewer Nsvs (1/2)**
>
> Thanks for the feedback. We appreciate the reviewer’s effort in evaluating our work and address each concern in detail below.
>
> ---
>
> > $\color{brown}{Question 1:}$ **Lack of comparison with the latest methods**: The paper fails to compare with the latest novel view synthesis methods such as ReconDreamer, FreeVS, and Dist - 4D. By comparing with these methods, the performance, advantages, and disadvantages of the proposed method can be more comprehensively evaluated, providing more valuable reference for readers.
>
> We acknowledge that ReconDreamer, FreeVS, and Dist-4D leverage generative models to perform view completion to assist scene reconstruction. However, the task of multi-traversal reconstruction considered in MTGS is different from generative scene completion from a single traversal. Due to the differences in input settings, supervision, and benchmarks, it is difficult to conduct a strictly fair comparison between these approaches and ours on existing datasets.
>
> That said, we view these two lines of work as complementary rather than competing. As illustrated in the ReconDreamer [demo](https://github.com/GigaAI-research/ReconDreamer?tab=readme-ov-file#rendering-results-in-lane-shift--6m-novel-trajectory), when encountering large lane deviations or heavily occluded regions, generative models may produce inconsistent content across frames, leading to blurry or over-smoothed rendering. Incorporating additional constraints from multi-traversal data, as in MTGS, could help mitigate such inconsistencies and potentially yield more stable and accurate reconstructions.
>
> ---
>
> > $\color{brown}{Question 2:}$ **Absence of novel view synthesis visualization**: The paper does not provide novel view synthesis visualization. It is recommended to visualize translations of 1m, 2m, and 4m. This kind of visualization can more intuitively show the effect of the method in novel view synthesis, helping readers better understand the performance of the method.
>
> We already provided novel-view synthesis visualizations in Figure 1, Figure 3, Figure A3, and Figure A4. As illustrated in Figure A1, all of these examples involve view extrapolations from the training trajectories with lateral offsets exceeding the lane width (approximately 4 meters). In Figure 1, translations of 2m, 4m and 8m are visualized. Some visualizations even include bird’s-eye-view synthesis (e.g., Figure 1 and Figure A3). We hope these results address your concern about the visualization of novel view synthesis.

---

> ### Author Response · Authors · 2025-11-25
> **Response to Reviewer Nsvs (2/2)**
>
> > $\color{brown}{Question 3:}$ **Poor - quality images**: The image quality in the paper is poor and cannot be used in real - world scenarios, and there is a large gap compared with the novel view synthesis of ReconDreamer. High - quality images are crucial for demonstrating the effectiveness of the method, and poor - quality images may affect readers' understanding and evaluation of the method.
>
> We respectfully disagree with this assessment. Visual appearance varies substantially across datasets due to differences in camera hardware, calibration quality, weather conditions, and scene complexity. In particular, the nuPlan dataset used by MTGS suffers from imperfect calibration and strong lens distortion, which inherently degrades the visual sharpness of all reconstructions on this dataset. As a result, it is not appropriate to directly compare "visual quality" between methods trained on different datasets.
>
> Quantitative metrics can offer a more objective reference, though they are still dataset-dependent. For example, our PSNR values are comparable to those reported by recent novel-view synthesis methods such as FreeVS, as shown in following table. Here we provide a quantitative comparison with FreeVS for clarity (note that this comparison is not strictly apple-to-apple and is intended only as a reference for *image quality*):
>
> | Method                                                | Training-View PSNR | Training-View SSIM | Novel-View PSNR | Novel-View SSIM |
> | ----------------------------------------------------- | ------------------ | ------------------ | --------------- | --------------- |
> | FreeVS (front-to-side, Waymo)                         | 24.96              | 0.730              | 20.70           | 0.628           |
> | **MTGS (unseen traversal with ~4m deviation, nuPlan)** | **28.73**\*        | **0.865**          | **21.58**       | **0.628**       |
>
> Based on these results, we would not characterize MTGS reconstructions as poor quality, and we kindly ask the reviewer to reconsider this judgment.
>
> ---
>
> > $\color{brown}{Question 4:}$ **Poor - quality reconstruction results in the demo**: The reconstruction results in the demo seem to be of poor quality, and there is still a large gap compared with the demo of OmniRe. The demo is an important means to show the practical effect of the method. Poor - quality reconstruction results may make readers question the practical value of the method.
>
> We again respectfully disagree. As discussed in our response to Question 3, it is not appropriate to directly compare reconstructions across datasets with different characteristics (e.g., Waymo versus nuPlan). A fairer comparison is to evaluate different methods on the *same* nuPlan data. If one directly compares MTGS’s nuPlan reconstructions with those of OmniRe on its official webpage (selecting the sixth dataset tab, nuPlan, for preview), MTGS produces visibly more faithful reconstructions.
>
> This difference arises from both the underlying datasets and the data preparation protocols. In our experiments, we apply the same data preparation to all methods for fairness, and MTGS consistently outperforms OmniRe on benchmarks. We therefore believe that MTGS provides strong practical value and kindly ask the reviewer to reconsider their assessment.
>
> ---
>
> > $\color{brown}{Question 5:}$ **Limited Innovation**
>
> We appreciate the reviewer’s concern regarding innovation. While MTGS utilizes several existing techniques, its core contributions lie in (1) introducing a dedicated reconstruction pipeline for repeated real-world traversals and (2) establishing a benchmark that explicitly advocates for the multi-traversal reconstruction paradigm.
>
> With the proposed three-node decomposition of multi-traversal scene graph, complemented by LiDAR‑guided exposure alignment, appearance‑residual modeling, and multi‑traversal consistency regularization, MTGS forms a coherent pipeline that enhances cross‑traversal fidelity, improves robustness to illumination shifts, and ensures stable novel‑view synthesis.
>
> Beyond the algorithmic design, MTGS provides a robust and reproducible data preparation and reconstruction pipeline for nuPlan, one of the most widely used end‑to‑end driving datasets. Our pipeline works even in the single‑traversal setting and significantly facilitates the construction of realistic simulation environments. The entire codebase will be released to support the research community.
>
> Thus, MTGS introduces both methodological and practical innovations, establishing a novel and meaningful paradigm for robust reconstruction under multi‑traversal setting.

---

### Official Review · Reviewer_buEE · 2025-10-29

**Soundness:** 3
**Presentation:** 3
**Contribution:** 2
**Rating:** 4
**Confidence:** 4

**Summary:**

In this paper, the authors propose a novel approach MTGS that reconstructs high-quality driving scenes from multi-traversal data. Specifically, MTGS employs a multi-traversal scene graph consisting of a static node and dynamic nodes. To address appearance variations across different traversals, the scene graph is further complemented by appearance nodes with learnable spherical
harmonics coefficient residuals. Experiments demonstrate that MTGS achieves state-of-the-art performance in both driving scene reconstruction and novel view synthesis.

**Strengths:**

1. This paper focuses on high-fidelity driving scene reconstruction using multi-traversal data, which is valuable for cross-lane simulations of AV. However, appearance variations in multi-traversal data disrupt scene consistency and introduce geometric errors, making the reconstruction process more challenging.
2. The authors decompose the entire driving scene using a multi-traversal scene graph with three core nodes: shared static nodes, appearance nodes, and transient nodes, which significantly improves the fidelity and geometric consistency of scene reconstruction.
3. Extensive experiments on the nuPlan dataset demonstrate that MTGS achieves superior performance.

**Weaknesses:**

1. One of the core innovations of MTGS—the "Scene Graph Node Decomposition"—has similarities to prior 3D Gaussian Splatting-related methods (such as StreetGS, DrivingGaussian) and lacks breakthrough ideas.
2. MTGS’s performance advantages are highly dependent on multi-traversal data, which poses significant drawbacks in real-world autonomous driving scenarios and hinders its practical deployment due to high data collection costs and inefficiencies.

**Questions:**

Please check the weaknesses.

---

> ### Author Response · Authors · 2025-11-25
> **Response to Reviewer buEE**
>
> Thanks for the valuable comments. We address your concerns below.
>
> ---
>
> > $\color{brown}{Question 1:}$ One of the core innovations of MTGS—the "Scene Graph Node Decomposition"—has similarities to prior 3D Gaussian Splatting-related methods (such as StreetGS, DrivingGaussian) and lacks breakthrough ideas.
>
> We acknowledge that some components build on existing techniques. However, we believe the proposed setting and pipeline are fundamentally novel. While prior works utilize a scene graph, our multi-traversal scene graph introduces distinct node types -- shared static, appearance, and transient -- which are not present in earlier methods. The appearance node explicitly models misalignment across traversals, while the transient node decouples dynamic content from the shared static node. Combined with LiDAR-guided exposure alignment, MTGS establishes stable in-traversal appearance consistency, allowing the appearance node to more effectively learn *cross-traversal* appearance differences.
>
> Furthermore, the geometric regularization, and our carefully designed calibration pipeline collectively ensure consistent and reliable shared-background geometry. This stable geometric foundation is crucial; without it, the model can easily overfit to traversal-specific artifacts, preventing the appearance node from independently modeling true cross-traversal variations.
>
> Taken together, the overall design of MTGS forms a technically novel framework specifically crafted to robustly capture traversal-dependent appearance variations and ultimately improve the fidelity of multi-traversal reconstruction with unique challenges.
>
> ---
>
> > $\color{brown}{Question 2:}$  MTGS’s performance advantages are highly dependent on multi-traversal data, which poses significant drawbacks in real-world autonomous driving scenarios and hinders its practical deployment due to high data collection costs and inefficiencies.
>
> We would like to clarify that multi-traversal data is already available in many real-world autonomous driving datasets, such as Argoverse 2, nuPlan, L2D, and the recently released PAI AV dataset, all of which contain repeated traversals of the same areas.
>
> In industrial settings, fleets routinely drive along fixed or frequently repeated routes, naturally accumulating multi-traversal data at scale. Such data can be efficiently queried for nearby traversals of a target scene and then exploited to build realistic reconstructions for simulation and downstream development. Therefore, while MTGS is indeed designed for multi-traversal data, we believe this requirement is well aligned with practical autonomous driving workflows.

---

### Official Review · Reviewer_9PmS · 2025-10-30

**Soundness:** 3
**Presentation:** 3
**Contribution:** 2
**Rating:** 4
**Confidence:** 4

**Summary:**

This paper proposes MTGS, a method for reconstructing driving scenes using multi-traversal data, where multiple recordings of the same road are available. The goal is to achieve high-quality view extrapolation and photorealistic driving scene reconstruction. To do so, the authors propose a multi-traversal scene graph with shared static nodes and traversal-specific appearance and transient nodes as welll as some tricks for better quality.  The authors conduct comprehensive experiments using the large-scale nuPlan dataset, which contains multi-traversal data.

**Strengths:**

- The paper is technically solid, well-organized, and supported by comprehensive ablation studies.
- The proposed framework effectively aligns appearances and reconstructs static environments with high visual fidelity in drivable areas.
- Leveraging multi-traversal data for reconstruction is an important and underexplored direction for autonomous driving and digital twin simulation.

**Weaknesses:**

- The handling of transient and dynamic objects remains problematic. Since these objects vary across traversals, the current model cannot establish consistent geometry or appearance. While the approach has potential as a strong background reconstruction technique, it is still incomplete as a holistic driving-scene solution.

- The method’s technical novelty is moderate. While the multi-traversal setting is valuable, the main modules (scene graph design, affine correction, normal/depth regularization) extend prior work rather than introducing fundamentally new formulations. The problem formulation is also very similar to reconstruction methods using in the wild images.

- As can be seen from the videos in the supplementary materials,  roadside vehicles are almost broken (in traversal_test).

- In Table 1, for ST setting, StreetSG and OmniRe are even worse than the original 3DGS in terms of PSNR, which needs better explanations.

**Questions:**

See weaknesses.

---

> ### Author Response · Authors · 2025-11-25
> **Response to Reviewer 9PmS**
>
> Thanks for the valuable review. We address each concern in detail below.
>
> > $\color{brown}{Question 1:}$ The handling of transient and dynamic objects remains problematic. Since these objects vary across traversals, the current model cannot establish consistent geometry or appearance. While the approach has potential as a strong background reconstruction technique, it is still incomplete as a holistic driving-scene solution.
>
> Thanks for highlighting MTGS's strength in background reconstruction. As stated in the limitation section (L481–L482), *"MTGS does not explicitly address the extrapolated views of transient objects"*. Integrating MTGS with complementary techniques, such as foreground object insertion or generative view completion, is a promising direction to further address this challenge, which we plan to pursue as future work.
>
> ---
>
> > $\color{brown}{Question 2:}$The method’s technical novelty is moderate. While the multi-traversal setting is valuable, the main modules (scene graph design, affine correction, normal/depth regularization) extend prior work rather than introducing fundamentally new formulations. The problem formulation is also very similar to reconstruction methods using in the wild images.
>
> We acknowledge that some components build on existing techniques, as we discussed in related works L112-124. However, the key technical novelty lies in extending dynamic scene graph designs to a **multi-traversal** setting. MTGS introduces three node types, shared static, appearance, and transient, that jointly address traversal-wise misalignment and dynamic content. This structured decomposition is not present in prior work.
>
> Regarding in-the-wild reconstruction, we discuss relevant methods in L141–L151 and compare with WildGaussians (Kulhanek et al., 2024). Our experiments show that directly applying in-the-wild reconstruction techniques (Tab. 2 ID-6) is insufficient to effectively solve the multi-traversal reconstruction problem, highlighting the necessity of our formulation.
>
> ---
>
> > $\color{brown}{Question 3:}$ As can be seen from the videos in the supplementary materials, roadside vehicles are almost broken (in traversal_test).
>
> As noted in the limitation section, *"In the shared background, there might be floating artifacts in regions not observed during training traversals, such as the space below parked cars."* These roadside regions are occupied by different parked vehicles across traversals, and many of these space are unseen during training. These floaring artifacts are unseen after applying transient nodes from any training traversal. If needed, it could be easily removed with 3D bounding boxes of parked cars.
>
> We would also like to emphasize that this evaluation and visualization use a *test-only traversal* to assess novel-view extrapolation performance. In real deployment scenarios, all available traversals can be incorporated during training, providing maximal coverage and minimizing unseen areas. During downstream applications such as simulation under a specific traversal, dynamic nodes would be applied, and these artifacts would not be visible.
>
> ---
>
> > $\color{brown}{Question 4:}$ In Table 1, for ST setting, StreetSG and OmniRe are even worse than the original 3DGS in terms of PSNR, which needs better explanations.
>
> Without geometric constraints from LiDAR depth supervision, 3DGS can more easily overfit to the pixel values of the training images under imperfect camera pose, leading to higher PSNR / SSIM but substantially worse on DINOv2 feature similarity and AbsRel metrics on novel-view. In contrast, StreetGS and OmniRe incorporate depth and geometric regularization, which improves geometric consistency at the cost of slightly lower PSNR in this setting.

---

### Official Review · Reviewer_aCbS · 2025-11-01

**Soundness:** 3
**Presentation:** 3
**Contribution:** 2
**Rating:** 4
**Confidence:** 4

**Summary:**

This paper presents a method for reconstructing realistic driving scenes from multiple trips along the same route. It separates static geometry from dynamic elements and adjusts for lighting or appearance changes, leading to cleaner, more consistent results. Experiments show that MTGS achieves noticeably better image quality and geometry accuracy compared to single-traversal methods.

**Strengths:**

1. The paper is well written and easy to follow.
2. The paper contributes to an interesting and important problem of reconstructing a scene across multi-traversals. Multi-traversal data often involve significant temporal and appearance variations such as changes in illumination, weather, and dynamic objects, which make achieving consistent, high-fidelity reconstruction difficult. This paper shows that their solution achieves strong results in multi-traversal reconstructions.

**Weaknesses:**

1. While I think this paper tackles an interesting topic. However, I think the most important challenge in the multi-traversal reconstruction is not enough handled. The decomposition of static and dynamic objects is common in self-driving gaussian splatting. While it is a natural solution in the multi-traversal scenarios, I believe it is not a significant contributions here. I feel the solutions to handle the illumination, weathers, etc. need to be strengthened in this paper. \
2. This paper seems lacking of visual comparisons, can the author provide more results comparisons on novel-view synthesis?

**Questions:**

1.I understand that the traversal-specific residual coefficients are meant to capture appearance differences such as lighting and reflections. However, since these residuals are learned independently for each traversal, wouldn’t that make it difficult for the model to generalize or interpolate lighting changes between traversals? In particular, because higher-order SHs (which normally encode directional illumination) are now traversal-specific, isn’t there a risk that the model just memorizes traversal-specific lighting instead of learning a shared representation of how lighting varies? From table 2, it seems without the Appr.Node, the novel-view psnr/ssim shows better results. \
2. You mentioned that sharing Y_0,0 forces the Gaussian to learn albedo. But since this is enforced only implicitly by sharing, without explicit supervision or intrinsic decomposition, how can you be sure that Y_0,0 doesn’t still capture some average lighting components?\
3. For the LiDAR-guided exposure alignment, view-dependent lighting effects such as shadows or specular highlights might be mistaken for exposure differences. Could this cause the method to over-correct and distort the true scene appearance?

---

> ### Author Response · Authors · 2025-11-25
> **Response to Reviewer aCbS (1/2)**
>
> Thanks for the insightful comments. We have carefully considered each point and respond as follows.
>
> > $\color{brown}{Question 1:}$ While I think this paper tackles an interesting topic. However, I think the most important challenge in the multi-traversal reconstruction is not enough handled. The decomposition of static and dynamic objects is common in self-driving gaussian splatting. While it is a natural solution in the multi-traversal scenarios, I believe it is not a significant contributions here. I feel the solutions to handle the illumination, weathers, etc. need to be strengthened in this paper.
>
> We appreciate the reviewer’s perspective and agree that multi-traversal reconstruction poses challenges beyond static-dynamic decomposition. While such decomposition is indeed used in prior works, MTGS goes beyond this by explicitly modeling *appearance variation* across traversals, which is a central difficulty in multi-traversal settings.
>
> Our formulation introduces an **appearance node** that captures traversal-dependent exposure, illumination, and color inconsistencies, which are key factors preventing naive multi-traversal fusion. Combined with **LiDAR-guided exposure alignment**, MTGS establishes stable in-traversal appearance consistency, allowing the appearance node to more effectively learn *cross-traversal* appearance differences.
>
> Furthermore, the **transient node**, **geometric regularization**, and our carefully designed calibration pipeline collectively ensure consistent and reliable shared-background geometry. This stable geometric foundation is crucial; without it, the model can easily overfit to traversal-specific artifacts, preventing the appearance node from independently modeling true cross-traversal variations.
>
> Taken together, the overall design of MTGS forms a cohesive framework specifically crafted to robustly capture traversal-dependent appearance variations and ultimately improve the fidelity of multi-traversal reconstruction.
>
> We will also improve the writing and overall logical flow of the manuscript to further strengthen these points. Thank you for your helpful comments.
>
> ---
>
> > $\color{brown}{Question 2:}$ This paper seems lacking of visual comparisons, can the author provide more results comparisons on novel-view synthesis?
>
> We have provided visual comparisions for multi-traversal reconstruction among 3DGS, OmniRE and MTGS in Figure 3, covering both training views and novel views. In addition, the supplementary material includes a video comparison between multi-traversal and single-traversal reconstruction on novel views. These results show that under the multi-traversal setting, MTGS effectively handles illumination differences and delivers clearer, sharper renderings for both training and novel views.
>
> ---
>
> > $\color{brown}{Question 3:}$ I understand that the traversal-specific residual coefficients are meant to capture appearance differences such as lighting and reflections. However, since these residuals are learned independently for each traversal, wouldn’t that make it difficult for the model to generalize or interpolate lighting changes between traversals? In particular, because higher-order SHs (which normally encode directional illumination) are now traversal-specific, isn’t there a risk that the model just memorizes traversal-specific lighting instead of learning a shared representation of how lighting varies? From table 2, it seems without the Appr.Node, the novel-view psnr/ssim shows better results.
>
> Yes, we agree that traversal-specific high-order SH coefficients can become view-dependent or even implicitly position-dependent in some regions (e.g., the nearby ground surface), as traversals are typically distributed in parallel along the road. Direct interpolation of high-order SHs across traversals may therefore be imperfect. However, because SH features are inherently interpretable, interpolating these coefficients can still produce a smooth and meaningful transition in appearance.
>
> Importantly, the goal of MTGS in the autonomous driving setting is **not** to learn a continuous or interpolable lighting space across traversals. Instead, MTGS is explicitly designed to **reconstruct and preserve traversal-specific lighting**, which is crucial for simulation-oriented applications. Downstream simulators require stable, traversal-consistent visual appearance when rendering extrapolated views along a given traversal. In such scenarios, memorizing traversal-specific illumination is desirable rather than a limitation. We have revised to clarify the slightly psnr/ssim drop in L451-454.
>
> We acknowledge that learning a unified or interpolable lighting embedding capable of modifying illumination or weather conditions within the same reconstruction remains an open challenge. Achieving such control would require additional supervision and mechanisms to ensure foreground–background consistency under lighting changes. We consider this a promising direction for future work.

---

> ### Author Response · Authors · 2025-11-25
> **Response to Reviewer aCbS (2/2)**
>
> > $\color{brown}{Question 4:}$ You mentioned that sharing Y_0,0 forces the Gaussian to learn albedo. But since this is enforced only implicitly by sharing, without explicit supervision or intrinsic decomposition, how can you be sure that Y_0,0 doesn’t still capture some average lighting components?
>
> We agree with the reviewer that sharing only the $Y_{0,0}$ coefficient does not guarantee a strictly "pure" albedo in the sense of intrinsic decomposition. As we stated in L233, *"forcing the Gaussian to learn its natural color and figure out commons in various appearances across traversals."* Our intention is not to claim that $Y_{0,0}$ becomes a physically perfect albedo map, but rather that, within the constraints of a radiance-field representation and multi-traversal supervision, sharing $Y_{0,0}$ is the statistically stable solution to explain appearance across traversals.
>
> Concretely, traversal-specific effects (such as lighting intensity, color tone shifts, reflections, and cast shadows) vary across traversals but remain relatively stable within each traversal. Because these effects are *not* consistent across traversals, the optimization cannot place them in the shared $Y_{0,0}$ channel; doing so would increase the reconstruction loss on other traversals. Consequently, such variations are absorbed by the traversal-specific residual SH coefficients $\beta^{\mathrm{residual}}_{i,T}$, which are designed to capture view-dependent and traversal-dependent components.
>
> Thus, $Y_{0,0}$ does not necessarily converge to a physically intrinsic albedo, but it converges to the maximally *traversal-invariant* component of appearance, which empirically resembles the natural color and stabilizes background geometry alignment. This is the intended meaning of "learning natural color and figure out commons" in our paper.
>
> We have revised the manuscript to clarify: *"This encourages each Gaussian to encode the appearance component that remains consistent among all traversals in $Y_{0,0}$, effectively capturing its traversal-invariant color and enabling stable background geometry alignment."*
>
> ---
>
> > $\color{brown}{Question 5:}$ For the LiDAR-guided exposure alignment, view-dependent lighting effects such as shadows or specular highlights might be mistaken for exposure differences. Could this cause the method to over-correct and distort the true scene appearance?
>
> Thanks for the insightful question. We agree that view-dependent effects such as shadows, reflections, or specular highlights, could in principle introduce bias when estimating exposure differences. In practice, however, the LiDAR-guided alignment remains stable for several reasons.
>
> - First, the on-board cameras are mounted close to one another relative to the distances to observed surfaces, which limits the magnitude of view-dependent appearance changes across cameras.
> - Second, the overlapping field of view between adjacent cameras is relatively large, providing a substantial number of LiDAR-anchored correspondences. Because our exposure estimation averages over all overlapping samples, outliers caused by highlights or shadows have minimal influence.
> - Third, much of the shared region lies on the ground plane, which exhibits weak view-dependent variation and thus offers a reliable photometric anchor.
>
> These factors collectively make the LiDAR-guided exposure adjustment robust in autonomous-driving scenarios. Additionally, our system still includes learnable per-camera affine transforms, which serves as a safeguard if any residual misalignment persists.
>
> We have added this clarification to Appendix A.2 to provide the relevant insight. Thank you again for your thoughtful question.

---

### Author Response · Authors · 2025-12-04
**General Author Response for Rebuttal**

Dear AC,

Thank you for handling our submission and for the reviewers’ thoughtful evaluations.

Our paper introduces **MTGS**, a multi-traversal Gaussian splatting framework that reconstructs high-fidelity driving scenes with multi-traversal driving logs, by jointly modeling shared static geometry, traversal-specific appearance variations, and transient objects. Reviewers consistently recognized the importance of the multi-traversal problem and acknowledged the clarity of our method and experiments.

Across the reviews, we observe a strong shared theme: **the methodology is sound, the problem is important, and concerns center primarily on requests for further justification, clarification, or additional qualitative results**, rather than on technical flaws.

Below we summarize the key strengths noted by reviewers and how our rebuttal directly addresses all major concerns.

---

**Recognized Strengths**

- **Clear problem motivation and well-organized method** `aCbS, 9PmS, buEE, Nsvs`.
- **Meaningful and underexplored direction**: multi-traversal reconstruction for autonomous driving `aCbS, 9PmS`.
- **Effective decomposition of scenes and strong reconstruction of static environments** `aCbS, 9PmS, buEE`.
- **Comprehensive experiments and ablations** `9PmS, Nsvs`.

---

**Addressing Key Concerns**

1. **Novelty and Relation to Prior Dynamic/Street-Scene Works** `aCbS(Q1), 9PmS(Q2), buEE(Q1)`

   - We clarified that MTGS is not a direct adaptation of prior scene graph methods as we discussed in related works.

   - The **multi-traversal scene graph** introduces *three distinct node types*, static, appearance, transient, that jointly address traversal-wise misalignment and dynamic content, a formulation not considered in prior works.
   - The appearance node with our **LiDAR-guided exposure calibration**, **geometric regularization**  form a unified pipeline that specifically targets cross-traversal inconsistencies, a unique challenge absent in single-traversal or in-the-wild settings.


2. **Comparison to Recent Generative NVS Methods**  `Nsvs(Q1)`

   - We explained that ReconDreamer, FreeVS, and Dist-4D operate in a different problem setting: single-traversal, generative view completion, and different benchmarks. Direct comparison is not meaningful.

   - We also note that MTGS and generative methods are *complementary*.


3. **Quality of Visualizations / “Poor Image Quality” Concern** `Nsvs(Q3,Q4)`

   - We clarified that nuPlan’s camera distortions and calibration challenges inherently limit visual sharpness compared to datasets like Waymo (used by OmniRe or ReconDreamer demos).

   - When comparing *on the same nuPlan data*, MTGS outperforms OmniRe both quantitatively and visually.

   - We provided a quantitative reference showing MTGS’s PSNR is comparable or better than generative NVS method FreeVS under similar conditions.


4. **Lack of Novel-View Visualization**  `aCbS(Q2), Nsvs(Q2)`

   - We highlighted that the paper **already contains multiple novel-view synthesis examples**, with lateral deviations of 2–8 meters, including BEV renderings (Fig. 1, Fig. 3, Fig. A3, Fig. A4).

   - We clarified these in the rebuttal and improved references in the manuscript.


5. **Practicality & Data Requirements** `buEE(Q2)`

   - We clarified that multi-traversal data is already **standard in modern AV datasets** (nuPlan, Argoverse 2, L2D, PAI-AV) and is routinely collected by industry fleets.

   - Thus, MTGS aligns well with real-world usage and incurs little additional cost.


6. **Detailed Discussions**
   - Additionally, reviewer `aCbS` provides valuable comments (Q3, Q4, Q5) on the multi-traversal appearance node and LiDAR-guided exposure alignment. We believe we have addressed these concerns and revised the manuscript accordingly.

---

**Final Remarks**

Overall, reviewers found MTGS **sound, well-motivated, and clearly presented**, with concerns primarily centered on clarification and additional comparisons rather than on technical errors. Our rebuttal and revisions directly address all raised concerns, and we hope this helps clarify the contribution and significance of MTGS.

Thank you for your consideration.

The Authors

---

### Meta-Review · Area_Chair_wF5t · 2025-12-22

**Summary:**

All reviewers acknowledged that the multi-traversal setting for novel view synthesis in autonomous driving is meaningful. However, they expressed concerns regarding the level of innovation of the paper, and all reviewers assigned scores marginally below the acceptance threshold.

Reviewer aCbS  and Reviewer 9PmS  both pointed out that dynamic objects should be considered in the multi-traversal setting. Reviewer buEE  raised concerns that the proposed method relies heavily on repeated traversals, which may limit its practicality in real-world deployment. Reviewers aCbS  , 9PmS  , and Nsvs  considered the experimental results to be insufficient or unconvincing.

**Reviewer Concerns:**

The authors responded by emphasizing the novelty of their method, improving the experimental results, and providing further justification for the experimental design. However, these revisions and explanations did not appear sufficient to address the reviewers’ concerns.

**Reviewer Scores:**

As none of the reviewers continued to participate in the discussion after the rebuttal, I believe that the original scores should remain unchanged.

---

### Decision · Program_Chairs · 2026-01-26

Reject